# Synergistic and antagonistic impacts of suspended sediments and thermal stress on corals

Rebecca Fisher [1,2], Pia Bessell-Browne [1,3] & Ross Jones [1,2]

Understanding pressure pathways and their cumulative impacts is critical for developing effective environmental policy. For coral reefs, wide spread bleaching resulting from global warming is occurring concurrently with local pressures, such as increases in suspended sediments through coastal development. Here we examine the relative importance of suspended sediment pressure pathways for dredging impacts on corals and evidence for synergistic or antagonistic cumulative effects between suspended sediments and thermal stress. We show that low to moderate reductions in available light associated with dredging may lead to weak antagonistic (less than expected independently) cumulative effects. However, when sediment loads are high any reductions in mortality associated with reduced bleaching are outweighed by increased mortality associated with severe low light periods and high levels of sediment deposition and impacts become synergistic (greater than what would occur independently). The findings suggest efforts to assess global cumulative impacts need to consider how pressures interact to impact ecosystems, and that the cumulative outcome may vary across the range of realised pressure fields.

[1] Australian Institute of Marine Science (MO96), University of Western Australia, 35 Stirling Hwy, Crawley, WA 6009, Australia. [2] Western Australian Marine Science Institute (MO95), University of Western Australia, 35 Stirling Hwy, Crawley, WA 6009, Australia. [3] The Centre for Microscopy, Characterisation and Analysis (MO10), University of Western Australia, 35 Stirling Hwy, Crawley, WA 6009, Australia. Correspondence and requests for materials should be addressed to R.F. (email: r.fisher@aims.gov.au)

The world's coral reefs are under increasing pressure from a range of global and local threats[1,2], posing challenges for environmental management and regulation of these valuable ecosystems[3–5]. With nearly the entire ocean affected by multiple stressors simultaneously[2], an understanding of how they interact is critical to predicting their cumulative impacts. While most assessments of cumulative impact assume additive effects[6], stressors may in fact interact such that their combined impacts are greater (synergistic) or even less (antagonistic) than might be expected in isolation[7].

A key global pressure is climate change-induced increases in sea surface temperature, leading to recurrent mass coral bleaching[8,9], potentially transforming coral reef assemblages[10,11]. In addition, elevated suspended sediment concentrations (SSCs), through runoff[12,13], resuspension[14] and dredging activities[15] are an important local source of reef degradation in coastal waters. Suspended sediments can impact corals through three inter-related pressure pathways[16], including: interfering directly with heterotrophic feeding[17,18], attenuating light and reducing rates of algal symbiont photosynthesis[19], and increased sediment deposition leading to smothering[20,21].

Using data from a large-scale dredging project undertaken on the reefs around Barrow Island in north Western Australia, we examine the relative importance of the three suspended sediment pressure pathways, along with thermal stress and associated coral bleaching, in predicting coral mortality. The 530 day Barrow Island dredging project coincided with a mass coral bleaching event[22] associated with a week of unusually warm water temperatures in the region[23]. The dredging project included extensive water quality and coral health monitoring, before, during and after completion of the dredging activities, at locations covering a gradient of dredging-related exposure, from within a few hundred meters up to more than 30 km from the dredging activities. This monitoring resulted in one of the largest datasets ever collected on individual tagged coral colonies exposed to dredging activities, and included estimates of dredging-related pressure (suspended sediments, available light, and sediment deposition), thermal stress (water temperature), along with measurements of coral health (~2-weekly measurements of ~1500 individual coral heads) throughout an entire 1.5 year dredging campaign. The fact that the data set captures not only a complete gradient of dredging-related pressure, but also temporally spans a thermal stress event, provides a unique opportunity to look at the cumulative impacts of these two important stressors in situ, and examine evidence for synergistic or antagonistic cumulative effects.

The analyses show that, depending on the severity of their impact on benthic light and sedimentation, suspended sediments may have both negative and positive effects on corals during periods of thermal stress. Low-to-moderate reductions in available light from suspended sediments can reduce the incidence of coral bleaching, and may reduce overall coral mortality, particularly for branching corals. However, when sediment loads are high any reductions in bleaching incidence are outweighed by increased mortality associated with severe low light periods and high levels of sediment deposition. The outcome is that under low sediment loads the cumulative impact of suspended sediments and thermal stress may be less than expected (antagonistic), whereas at high sediment loads the overall impact is greater than when these stressors occur in isolation (synergistic). The results highlight that while pressures such as thermal stress associated with climate change can only be managed at a global scale, management of local pressures may, in some cases, have the capacity to modify their overall impact.

## Results

**Incidents of coral bleaching and mortality at Barrow Island.** Incidents of both whole and partial mortality of branching (Acropora spp. and Pocilloporidae) and massive (Poritidae) corals was observed as a result of both the combined impacts of dredging activity, and thermal bleaching (Fig. 1). For the branching corals whole coral mortality was 27% (65 of 241 of colonies dying) at the end of the project (after the combined impacts of both dredging and the bleaching event), with whole colony mortality distributed across sites both near and far from dredging (Fig. 1a). However, 26% of colonies (63 of 241) showing no mortality at all (Fig. 1a). For massive corals whole colony mortality was < 1% (1 of 400 of colonies dying) by the end of the project, but only 11% (45 of 400) of colonies showed no mortality at all (Fig. 1b), suggesting that small to moderate levels of partial mortality were common for this growth form. Whole colony mortality usually occurred across more than one fortnightly survey period, with only two individual partial mortality events between consecutive surveys representing complete loss of the colony (partial mortality event ~1, Fig. 1c, black open circles). Site mean partial mortality values were quite variable, ranging from near zero to up to 3 to 4% of the colony surface per fortnight, but were generally highest at sites nearest to dredging (Fig. 1c, d, red line). The highest proportion of colonies showing bleaching (>30%) were associated with warmer temperatures and were generally at sites father from the dredging activities, although moderate and low levels of bleaching (<30%) occurred across sites both near and far from dredging (Fig. 1e, f).

**Relative importance of mortality pathways.** Both our structured equation and full subsets analyses exploring suspended sediment and thermal pressure metrics and stress indicators showed that the mortality of branching and massive corals was driven by both temperature and bleaching, as well as dredging-related pressures (Fig. 2). This highlights the complex interaction between dredging and thermal stress that occurred during the dredging project. There was little evidence that SSC per se was a strong predictor of mortality, showing a low total standardized effect size (Fig. 2a) and low summed AICc weight values (Fig. 2b, c). Thus, while SSC is directly responsible for generating both low light and high sediment deposition stress, it is not an important pressure independently, providing strong field support for the findings of recent laboratory studies[24]. Loss of benthic light was the strongest predictor of the incidence of partial mortality in branching corals (Fig. 2b) and was also important in predicting the amount of live tissue loss in massive corals (Fig. 2c). Periods of low light and darkness limit photosynthesis, which is considered detrimental as symbiont photosynthesis can provide corals with up to 90% of their daily energy requirements[25,26]. Reduction in phototrophic energy generation potentially increases reliance on energy stores, which would gradually decline over time, resulting in mortality once sufficiently depleted[27]. Sediment deposition was important in predicting partial mortality incidence in massive corals (Fig. 2b), but there was no evidence sediment deposition directly impacted branching corals (Fig. 2b). While coral colonies employ a range of active and passive removal mechanisms to clear sediment[28–30], when deposition rates exceed these removal mechanisms, more rapid necrosis of tissue results[21,31].

**Predicting bleaching of branching and massive corals.** The probability of bleaching was strongly driven by an interaction between temperature and light for both branching and massive corals (Supplementary Table 2, AICc weight = 1). Bleaching only occurred under conditions of high thermal stress (daily mean

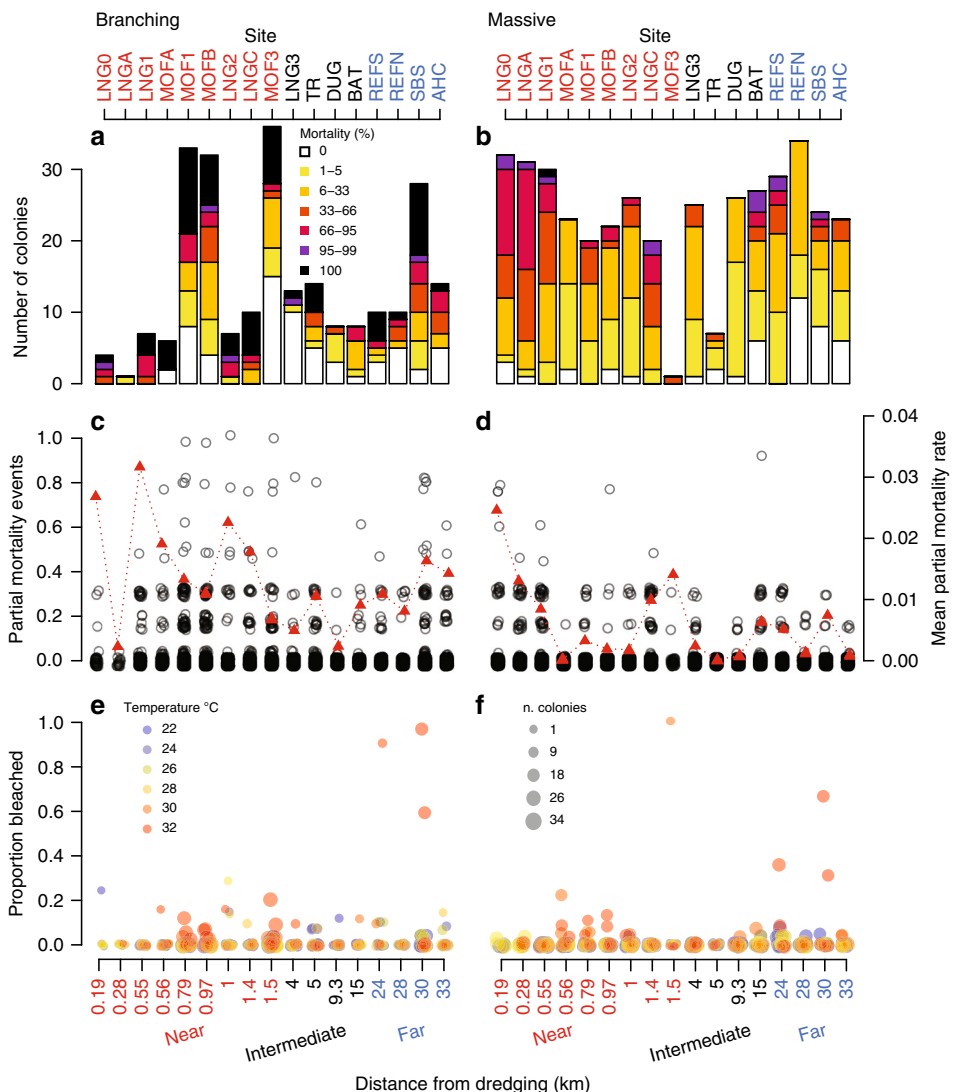

**Fig. 1** Mortality and bleaching incidence across 17 sites at Barrow Island. The frequency of branching (left hand panels) and massive (right hand panels) coral colonies exhibiting each of seven mortality scores at the end of the dredging project (**a**, **b**), the magnitude of individual partial mortality events between each survey (~1–3 weeks) (open black circles, **c**, **d**), mean partial mortality (red triangles, **c**, **d**); and the proportion of colonies exhibiting bleaching between each field survey (**e**, **f**), with colour representing the worst case temperature value over the survey period and size indicating the number of observed colonies. X-axis label colours indicate three categories of sites based on previously established patterns in the severity of water quality conditions, including: near (<1.5 km), intermediate (4–15 km) and far (24–32 km) from dredging

temperature > 29 °C) at high light values ( > ~4 mol photons m$^{-2}$ per day, Fig. 2a), a phenomenon that has been observed previously[32,33]. It appears that the reduced light levels associated with the suspended sediments generated during dredging may have alleviated the extent of bleaching, a finding consistent with recent laboratory[34] and field[35,36] studies showing lower incidence of bleaching on inshore and/or more turbid reefs and supporting the idea that such reefs may act as refuges under ocean warming[37]. While the interaction between light and temperature was evident for both types of corals, there was a higher overall probability of observing bleaching for branching than for massive corals (Fig. 3a, b), which is consistent with studies examining relative coral sensitivities to thermal stress[38].

**Predicting mortality of branching and massive corals**. The incidence of any partial mortality for branching corals was best predicted by an interaction between bleaching status, along with both temperature and light (Supplementary Table 2). Once a coral showed signs of thermal bleaching, the probability of observing

partial mortality in branching corals was high (around 40%; Fig. 3e). Both light and temperature strongly influenced the incidence of partial mortality events for branching corals, and this is particularly strong for corals showing no prior signs of bleaching (Fig. 3c, Supplementary Figure 4a). At high light levels ( > 1.9 mol photons m$^{-2}$ d$^{-1}$) the incidence of partial mortality increases to values of around 30–40% at the highest temperatures (>31.5 °C), but has no negative impact on corals in the absence of thermal stress (Fig. 3c). Conversely, across the range of observed temperatures low light conditions strongly increase the incidence of partial mortality for unbleached branching corals, with partial mortality events as frequent as 40% under extremely low light conditions (<0.24 mol photons m$^{-2}$ d$^{-1}$) at low temperature (22–28 °C), and as high as 60% at high temperature (>30 °C, Fig. 3c). Overall, the greatest incidence of mortality of branching corals (both bleached and unbleached) occurred under conditions of both high temperature stress and extremely low light (Fig. 3c, e). The findings support experimental studies suggesting that mortality risk from bleaching is closely associated with energy status in corals[34].

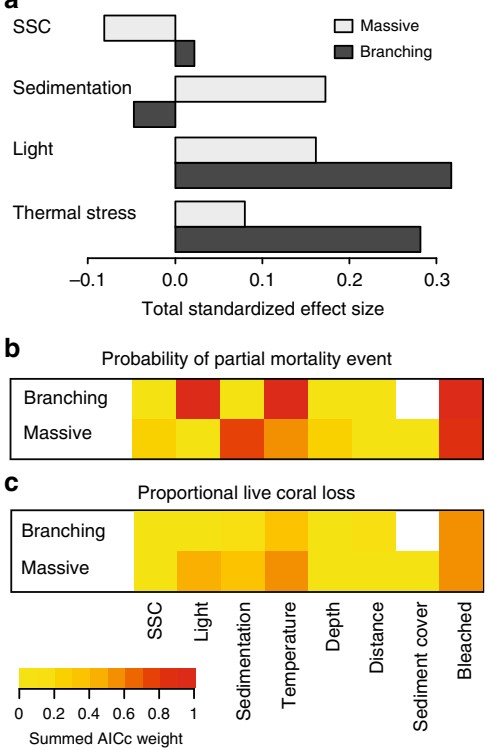

**Fig. 2** Relative importance of thermal stress and dredging mortality pathways. Total standardized effect size (**a**) based on summed path coefficients for suspended sediment concentrations (SSC), sedimentation, light, and thermal stress; and summed AICc weight values from a full subsets additive regression analysis for the probability of observing any partial mortality (**b**) and proportional live tissue loss (**c**, Methods 'Predicting coral mortality')

The incidence of partial mortality events for massive corals was higher in bleached corals, but also increased with sedimentation in both bleached and unbleached corals (Fig. 3d, f; Supplementary Figure 4b). There was also evidence that the incidence of any partial mortality also increased with temperature (Fig. 3d, f) and only limited support for any interaction among predictors (Supplementary Table 2). The probability of any partial mortality occurring increased by about 40% from the lowest to highest levels of sedimentation, and was on average about 20% higher for corals showing evidence of thermal bleaching (Fig. 3d, f).

When partial mortality events did occur, the proportion of live tissue lost was only weakly predicted by the pressure metrics examined (Fig. 2c, Supplementary Table 2). Bleaching appeared to have some effect, with the amount of live tissue lost about 30% higher for corals showing evidence of thermal bleaching in both branching and massive corals (Fig. 3g, h). For massive corals the proportion of tissue loss also increased with decreasing light levels (Fig. 3h) and/or sedimentation (Supplementary Figure 4d). Overall, for branching corals the results suggest a high level of mortality associated with thermal bleaching that is further exacerbated by dredging-related stress. Bleached massive corals, on the other hand, only show levels of mortality similar to that of branching corals whilst also under conditions of high dredging-related pressure. Levels of mortality of thermally bleached corals depend on the severity of thermal stress[10]. The lower resilience of bleached massive corals under conditions of high sedimentation and low light likely reflect the reduced capacity for bleached corals to remove sediments[39].

**Cumulative impacts of suspended sediments and thermal stress**. Using a Monte–Carlo simulation approach based on Bayesian fits of the best models we predicted fortnightly partial mortality under the following three scenarios: (1) thermal pressure; (2) dredging pressure (black lines, Fig. 4a, b); and (3) cumulative thermal and dredging pressure (red lines, Fig. 4a, b). From the thermal pressure alone and dredging pressure alone scenarios we also calculated a theoretical 'additive' prediction (blue lines, Fig. 4a, b; see "Methods" section 'Estimating individual and cumulative impacts'). For branching corals, there are a range of light values where the cumulative impacts on coral appear to be below the theoretical additive level (red line is below the blue line, Fig. 4a) and appear slightly antagonistic (i.e., less than in isolation, Fig. 4c). This weak antagonistic effect reflects the reduced incidence of bleaching associated with the shading effect of low light conditions during dredging. Peak probability of antagonistic cumulative impacts occurs at 3.5 mol photons $m^{-2}$ per day for branching corals, and indicates the optimal light threshold for minimising thermal stress. However, at 1.2 mol photons $m^{-2}$ per day simulated cumulative impacts on coral loss cross the theoretical additive level (red line is above the blue line, Fig. 4a), indicating the impacts are now synergistic (i.e., worse than in isolation, Fig. 4c). This occurs because elevated mortality associated with very low light conditions (and presumably the associated reduced energy status[34]) exceeds any gain associated with reduced incidence of bleaching.

For massive corals, the reduced coral loss through potential shading effects is less than for branching corals, with limited support for antagonistic effects at any light values (Fig. 4b, d). This occurs because high sedimentation and low-light conditions increase the incidence of mortality as well as the amount of coral tissue lost in both bleached and unbleached massive corals, and bleached massive corals do not necessarily have a high probability of mortality when dredging conditions are benign. Both reduced light[34] and increased sedimentation are likely to decrease the energy status of corals, particularly when the suspended sediments themselves infer no nutritional value, as in the case when dredging material of very low organic content, such as occurred at Barrow Island. For massive corals, the transition from antagonistic to synergistic cumulative impacts occurs at 3.1 mol photons $m^{-2}$ $d^{-1}$. While this intersection point occurs at a relatively high light value for massive corals (this value is naturally exceeded in the absence of dredging during winter months, see Supplementary Figure 2a), the transition to synergistic cumulative impacts is very gradual (Fig. 4d), with an 80% probability of synergistic impact not reached until much lower light levels (i.e., much less than the 1.2 mol photons $m^{-2}$ per d threshold for branching corals, Fig. 4a).

## Discussion

At high dredging-related pressure, our models suggest strong evidence for synergistic impacts between sediment-related stress from dredging and thermal stress on corals, with overall mortality highest when these impacts occur together. Depending on the sources of the suspended sediments, expected synergistic impacts may only occur over very limited spatial scales. We estimated partial mortality as a function of distance from the dredging activities using the fortnightly water quality conditions observed at Barrow Island (Fig. 4e, f, Methods 'Distance of impacts'). Estimated distances of 50% effect on mortality under thermal stress occurred at 0.45 km (0.24–2.5 km) for the branching corals (Fig. 4e) and 1.0 km (0.19–8.1 km) for massive corals (Fig. 4f). In the absence of thermal stress distances of 50% effect on mortality were 0.92 km (0.19–4.3 km) for branching corals (Fig. 4e) and 0.29 km (0.19–0.50 km) for massive corals (Fig. 4f). This suggests

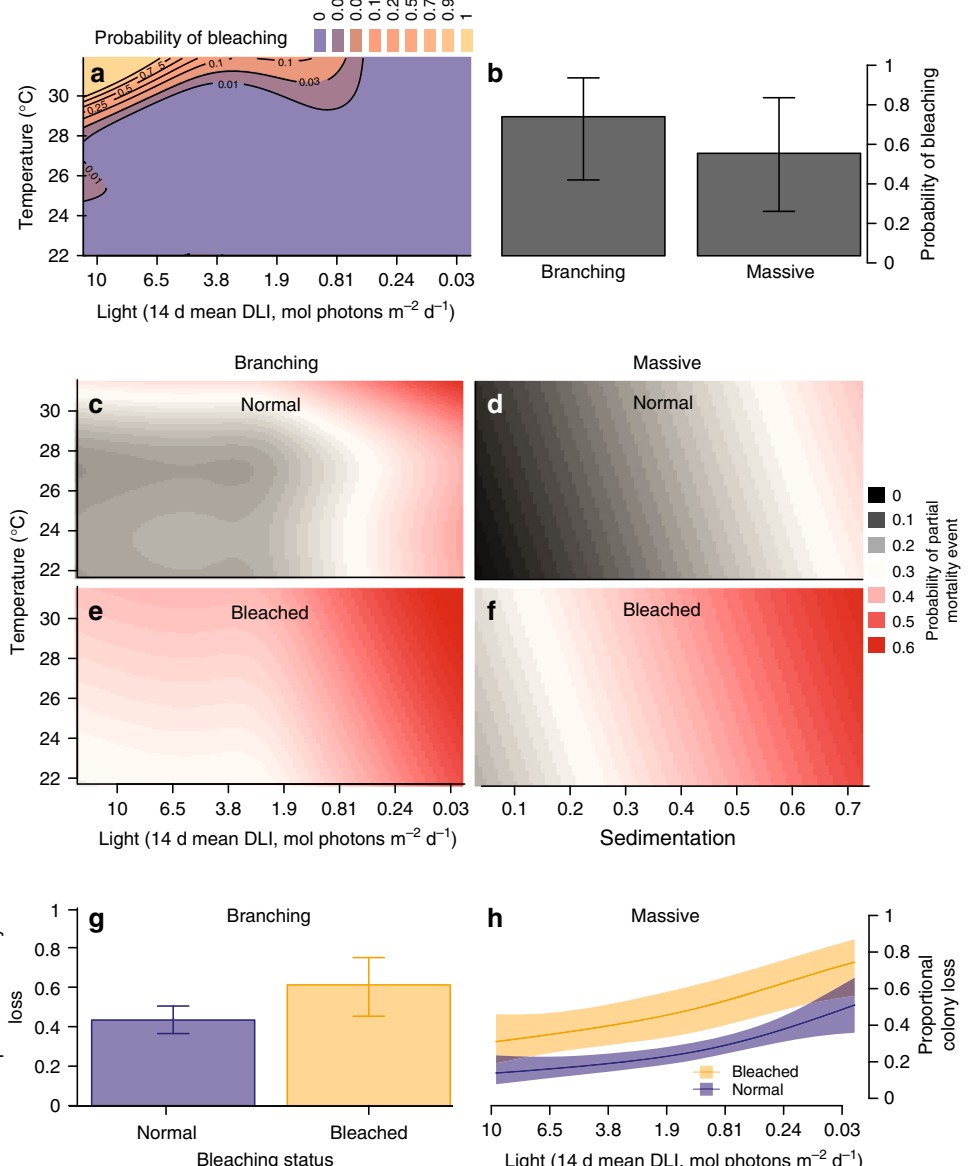

**Fig. 3** Relationships between thermal and dredging-related stress metrics and coral health. Mean predictions from the best fit models (lowest AICc, Methods 'Predicting coral mortality') for the probability of bleaching (**a**, **b**), probability of mortality (**c**–**f**), and the proportional loss of coral tissue given mortality (**g**, **h**). Error bars are 2 × estimated se, as reported from the gamm4 model fit

that while cumulative synergistic impacts may increase the distances of expected mortality of massive corals, overall most mortality is occurring within a few hundred meters from the dredging activity.

Our results highlight that the outcome of concurrent pressures is not always additive, and in fact may change from antagonistic to synergistic across realised pressure fields, potentially complicating efforts to assess global cumulative impacts[6,40]. While pressures such as thermal stress associated with climate change can only be managed at a global scale, management of local pressures may, in some cases, have the capacity to modify their overall impact. A clear understanding of how locally manageable pressures interact with global and regional pressures (and via which pathways) is critical for effective management. Evidence for a shading effect associated with reduced light lends support to the idea that it may be possible to alleviate the coral loss associated with thermal stress through active mitigation efforts[41], at least at local scales.

## Methods

**Study details**. From 19 May 2010 to 31 Oct 2011 (530 days), a large-scale capital dredging project was undertaken on the reefs around Barrow Island located ~50 km offshore the Pilbara region of NW Western Australia (Supplementary Figure 1) to provide ship access to a liquefied natural gas (LNG) processing plant on the Island. The project used a mix of trailer suction hopper, cutter suction and backhoe dredges, removing an estimated 7.6 Mm³ of predominantly unconsolidated, undisturbed carbonate sediments overlying limestone pavement[42]. It was predicted during the environmental impact assessment processes that the dredging campaign would have some effect on corals, and these impacts were permitted to occur under state and federal approval conditions. The project was carried out under a rigorous environmental management plan[43].

The project involved collection of extensive time series of both water quality and coral health monitoring across 25 sites covering a gradient from near ( < 1 km) to = far (>30 km) from the dredging activities[42]. Water quality and coral health monitoring data were collected throughout the duration of the project, which also included a period of widespread thermal bleaching which occurred in the summer of 2010–2011 due to a warm water anomaly[22,44], providing a unique opportunity to explore the cumulative impacts of suspended sediment loads and thermal stress on coral reefs. Access to associated monitoring datasets, including water quality and coral colony images throughout the dredging project were obtained through the WAMSI Dredging Science Node (https://www.wamsi.org.au/dredging-science-node).

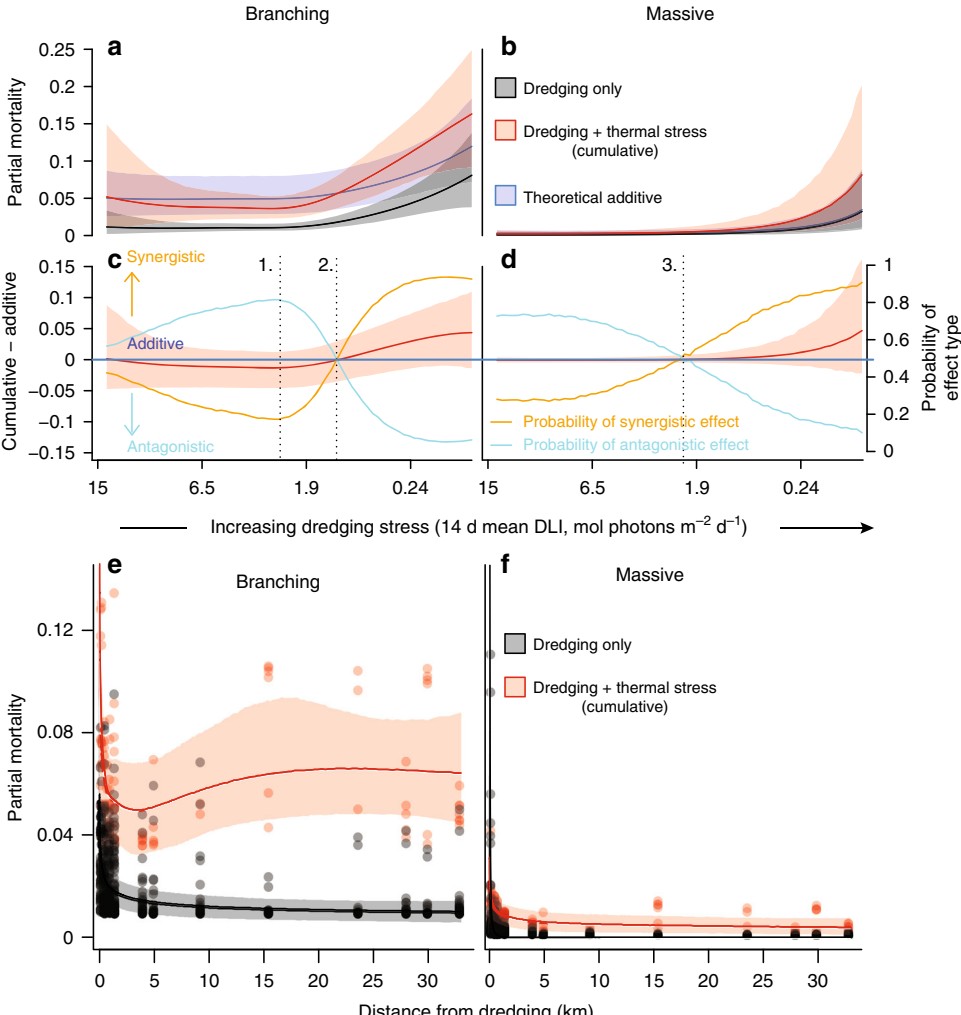

**Fig. 4** Cumulative impacts of suspended sediments and thermal stress. Fortnightly partial mortality (total proportional coral loss) was calculated across a gradient of increasing dredging-related stress (decreasing 14 d running mean light values) for both branching (**a**) and massive corals (**b**). The red lines indicate the cumulative effect of dredging pressure and thermal stress, whereas the black lines indicate the effect of dredging alone. The blue line represents a theoretical "additive" curve, estimated as the sum of coral loss associated with thermal stress alone (no dredging stress and high light levels) and that predicted for different levels of dredging stress at low temperature. Shaded bands represent the 95% confidence bounds of the Monte–Carlo simulation. Additive impacts were subtracted from the cumulative impacts to determine if impacts were synergistic (greater than expected independently) or antagonistic (less than expected independently, **c**, **d**, Methods 'Estimating individual and cumulative impacts'). Fortnightly partial mortality was also examined against distance to dredging activities (**e**, **f**, Methods 'Distance of impacts') under both high (31 °C, red bands) and low (25 °C, black bands) thermal stress conditions. Three light based thresholds were derived, including: 1. the maximum probability of antagonistic impacts, 2. the intersection between antagonistic impacts and synergistic impacts for branching corals, and 3. the intersection between antagonistic impacts and synergistic impacts for massive corals

Before the start of dredging ~25 water quality and coral health (see below) monitoring sites ranging in depth from 4 to 11 m were installed from a few hundred metres to several tens of kilometres away from the future dredging activities (Supplementary Figure 1). Sites were installed from June 2009 onwards and the installations were complete by the start of the dredging on May 20, 2010. During dredging there was a near continuous unidirectional, southerly movement of the sediment plumes throughout the 1.5 y study (see ref. [45,46]). For the purposes of this study, analysis of water quality and coral health was restricted to 17 sites to the south of the main dredging and 2 reference sites (AHC and REFN) located > 28 km to north of the study area (Supplementary Figure 1). The sites were grouped into 3 categories based on previously established patterns in the severity of water quality conditions[46], including: near dredging ( < 1.5 km), intermediate distances from dredging (4–15 km), and far from dredging (24–32 km), encompassing sites both to the north and south of the primary dredging site (Supplementary Figure 1).

**Water quality monitoring**. Water quality measurements were made at 10 min intervals for the pre-dredging and dredging phases using instruments attached to a seabed mounted steel frame. Turbidity was measured using sideways facing optical fibre backscatter (OBS) nephelometers while photosynthetically active radiation (PAR) was recorded using $2\pi$ quantum sensors. For all turbidity data, any values < 0 NTU were removed, and a smoothing filter was applied where any value > 3 NTU if the value was more than 2.5 × the mean of the preceding and following value[42]. Benthic light was measured using a $2\Pi$ quantum sensor, with 10 min readings modelled using generalised additive models (GAM) for each day separately, with predicted values used to determine the sum of the per second quantum flux measurements[46,42] as the daily light integral (DLI) calculated as mol photons m$^{-2}$ d$^{-1}$. The instrument platform also included a sediment accumulation sensor[47,48] which, while unable to provide absolute units of sedimentation, provides a relative sedimentation index[49], and was scaled to between 0 and 1. Water temperature (degrees Celsius, °C) was recorded by the in situ loggers. All data were visually screened for anomalies and evidence of logger failure using time series plots of the raw data, and any suspect data were removed, with all removed data recorded in a data screening log[42].

**Coral health monitoring**. For each of the water quality monitoring sites, 50 coral colonies considered representative of the local reef community were selected for sequential monitoring and tagged. Additional colonies were also tagged to ensure a

minimum of 20 massive Poritidae (*Porites lobata* and *Porites lutea*, hereafter referred to as *Porites* spp.) colonies per site. Each colony was photographed in plan view (from above) using a digital camera, with a set of reference photographs taken at the start of the program and used to create a reference guide showing the original image of each colony to compare against for each additional survey, to ensure that the correct corals were photographed on each occasion, and that consecutive photographs were taken at the same orientation[50]. A metal frame was used to ensure all photographs were taken at the same distance from the substratum, and also provided a scale. The colonies were photographed every ~14 d for the 530 d duration of the dredging (until 11 November 2011) with 27–40 separate field surveys undertaken to photograph each individual tagged coral colony at each site over the duration of the project.

The growth form of each colony was recorded as either encrusting, foliose, corymbose, branching or massive. Consecutive photographs of each individual colony were scored for partial mortality, cover by sediment, and thermal bleaching, as a percentage of live tissue of the specific tagged colony, based on a relative assessment of the preceding photograph(s). Percentages were recorded on a categorical scale ranging from 1 to 7 where 1 = 0%, 2 = 1–5%, 3 = 6–33%, 4 = 34–65%, 5 = 66–95%, 6 = 96–99%, and 7 = 100% coverage, which were converted to their equivalent proportional cover midpoints (1 = 0, 2 = 0.03, 3 = 0.19, 4 = 0.495, 5 = 0.805, 6 = 0.970, and 7 = 1) and used as a continuous response variable in subsequent summaries and analyses. Because of the photographic time sequence, it was possible to follow the fate of the coral tissue in time, making it possible to clearly observe mortality and/or bleaching of tissue, and in many cases identify the cause (thermal or sediment related stress). Colonies often became partially covered in sediment and if sediments were ultimately washed off the coral surface (by waves or currents) revealing live tissue, the sediment covered tissue was classified as 'live' throughout the relevant period of the photograph sequence. If the sediments remained on the surface, or once washed off revealed a dead surface, the tissue was classified as 'dead' from when the sediment covering was first observed. The same principle was used in the case of mucous sheet formation in massive *Porites* spp.[51]. Similarly, the image time series allowed bleaching associated with sediment sitting on coral tissue and being subsequently washed off to be distinguished from other, non-sediment related bleaching (largely thermal stress related).

Where it was clear from the photographic time series that a colony had been moved or dislodged either by swells and waves associated with tropical storms (TC Bianca or TC Carlos, see Supplementary Figure 2), the colonies were excluded from all further analyses.

While a broad range of coral families and genera were included across the original data set as being 'representative of the community', in most cases there was insufficient replication at each site for meaningful analyses of individual families. Instead, we focus on two key groups: 'Massive' and 'Branching', representing two contrasting growth forms that likely differ in their susceptibility to both sedimentation and light stress. Because massive *Porites* spp. were a focal group for compliance monitoring during the dredging program (see above) they occurred in sufficient numbers across the full design to be analysed as a single group (400 colonies across 17 sites, median number per site per field survey = 23). While other massive colonies were also observed (for example from the Mussidae, Diploastreidae and Lobophylliidae families) these were not generally well represented across sites, and their inclusion may bias results given that family level differences can be much stronger predictors of bleaching susceptibility than overall growth form[52]. Thus for the purposes of this study the term 'Massive' refers specifically to massive and sub-massive Poritidae, most likely a mix of *Porites lobata* and *Porites lutea* (see above). Compared to the massive Poritidae, observations of branching corals within any one family were scarce and it was necessary to aggregate the branching and corymbose *Acropora* spp. with the Pocilloporidae (all of which were branching forms, 241 colonies across 17 sites, median number per site = 10). The genus *Acropora* was the most abundant branching genera across the data set, with good spatial representation. The Pocilloporidae included representatives from *Seriatopora* spp., *Stylophora* spp. and *Pocillopora* spp., all of which have relatively similar morphologies and thermal stress sensitivity to the branching/corymbose *Acropora* spp.[38,53], and were also well distributed across the study design. Studies including species from both the *Acropora* spp. and Pocilloporidae indicate similar sediment clearance capabilities[30] and similar responses to light reduction[19]. The only other (non-Poritidae) branching/corymbose corals in the data set were a small number of *Astreopora* spp. and *Hydnophora* spp., both occurring only at two sites. These two additional branching groups were not included as they differ markedly to the branching *Acropora* spp. and Pocilloporidae in their sensitivity to bleaching[38] and because of their poor spatial representation. While there are certainly some differences in the life history characteristics of our 'Branching' group (such as mode of reproduction) that might result in different rates of recovery from disturbances[54], for the purpose of exploring the relative effects of thermal and sediment related stress on adult massive versus branching corals, members of the 'Branching' group can reasonably be considered to be functionally similar.

**Predicting coral mortality**. To explore the relative importance of the different suspended sediment and thermal stress pathways for mortality of corals at Barrow Island, and to build predictive models of coral mortality, a range of environmental and coral parameters were derived and used in statistical analyses (Supplementary Table 1). This included: two stress indicators derived from coral health parameters collected as part of the image analysis ('Bleached' and 'Sediment cover'); three environmental parameters based on water quality logger data designed to capture the three stress pathways associated with the suspended sediment loads cause by the dredging activities ('SSC', 'light' and 'sedimentation, see[16]) (Supplementary Table 1); temperature (also based on site specific water quality logger data); and depth and distance to the dredging activity.

All the data were processed, screened and analysed in R (version 3.2.3[55]). We used two approaches to investigate which mortality pathways have most support based on this Barrow Island data set, for both the branching and massive coral groups. In the first approach we used path analysis implemented in a structural equation modelling framework, via the sem package[56,57] in R. We used the logit of the proportional mortality of corals between each image (live cover at time t – live cover at time t + 1) as the response, for the branching and massive coral groups separately (see 'Coral health monitoring' above), averaged at the site and field survey level (n = 473 field survey x site means for 'Branching' and n = 478 field survey × site means for 'Massive'). An a priori full model was developed for each coral group (see Supplementary Figure 3). For path analyses we were explicitly interested in the relative importance of the three suspended sediment pressure pathways (SSC, light and sedimentation, see[16]) as well as thermal stress indicators (captured in our path analysis as 'bleaching' and as elevated water temperature) in causing mortality of corals. The suspended sediment concentrations (SSC) could directly cause mortality, or could cause reductions in light availability and increased sedimentation. Reduced light availability could directly cause mortality. Sedimentation could cause partial mortality directly, or via sediment cover, which may subsequently result in mortality. As branching corals were never observed with sediment covering their surfaces, this link was not included in the branching coral partial mortality pathways. The total standardized effect of SSC, light and sedimentation was calculated by multiplying the standardized coefficients within a pathway then summing the coefficient values for all relevant pathways. The total standardized effect for thermal stress was the sum of the pathway coefficients via the effect of elevated temperature on bleaching, which in turn impacted the response, as well as the direct effect of elevated temperature (see Supplementary Figure 3).

In addition to the path analysis, we used a full-subsets modelling approach[58] to determine which of the available predictor variables were most important in driving partial mortality of branching and massive corals, and to establish the best statistical models for predicting partial mortality under thermal and suspended sediment related stress. In this approach a complete model set was constructed and subsequently compared using Akaike Information Criterion for small sample sizes (AICc) and AICc weight values (ω)[59]. The relative importance of each predictor variable was determined by summing the ω values for all models containing the variable, with higher summed values representing increased importance of that predictor to the response variable[59]. To avoid issues associated with collinearity between predictors, predictors were only included in a single candidate model when their absolute Pearson correlation coefficient was less than 0.4. To ensure models remained ecologically interpretable, only models with up to three predictors were included in any one candidate model (although all predictors were examined across the whole model set). We also investigated interactions between factors (Bleached and Sediment Cover) and continuous predictors where these were deemed appropriate.

To explicitly separate bleaching impacts from mortality, the probability (or incidence) of bleaching was examined as a function of light and temperature, with morphological group (branching or massive) included as a factor. Data were the number of colonies observed during a field survey showing a higher bleaching score than the previous survey (successes in a binomial call) modelled as a function of the number of observed colonies (trials in a binomial call, median number per site per field survey = 10 for 'Branching' and 23 for 'Massive'). A site identifier was included as a random effect to accommodate repeated sampling of the same sites over time.

As the observed partial mortality events across the time series were highly zero inflated for both morphological groups, we used a two-step hurdle approach to model mortality. Firstly, the probability of observing a partial mortality event (a colony photographed during a field survey showed a higher mortality score than in the previous survey) was modelled at the individual colony level, including colony ID as a random effect to accommodate repeated sampling of the same colonies over time (median number of surveyed colonies per site per field survey, as above: 10 for 'Branching' and 23 for 'Massive'). In the next step, only the data where partial mortality events were actually observed were included, and the amount of coral loss (live cover at time t – live cover at time t + 1) modelled as a function of the proportional live cover at time t. Colony ID was again included as a random effect. Sample size for the presence only model was obviously much lower than for the presence-absence model, as this was limited only to colonies showing some partial mortality during a field survey. There were 318 and 164 colony level observations across all sites and field surveys for the 'Branching' and 'Massive' group respectively.

Models were fitted using the gamm4 function in the gamm4 package[60], with the presence of mortality modelled as a binomial distribution. We used generalised additive mixed models rather than generalised linear mixed models to allow for potential non-linear relationships between the response variable and the various continuous environmental predictors. Smoothing terms were fit using a cubic

regression spline[61], with the 'k' argument limited to 5 (to reduce over-fitting and ensure ecologically interpretable monotonic relationships). To further reduce overfitting, the maximum number of predictors allowed in any given model was restricted to three. A null model consisting of only an intercept and the random factors was also included in the model set to test if any of the included variables were indeed useful predictors of coral mortality events. While the data modelled here are time series, temporal variograms did not show strong consistent trends in temporal autocorrelation that could be modelled effectively using the correlation structures available in mgcv. Overall, temporal dependence was not strong because only a single data point from the daily water quality time series were used from the time step between observations of individual coral heads. Inclusion of colony level ID random effects effectively deals with the non-independence associated with some colonies being simply more robust or more sensitive to mortality and bleaching.

**Estimating individual and cumulative impacts**. We fitted the best models identified through the full subsets modelling for predicting the incidence of bleaching, the incidence of mortality, and proportional coral loss (given mortality) in a Bayesian framework using the function stan_gamm4 from the rstanarm package[62] in R. During initial Bayesian fits, we used three chains with 10,000 iterations and the rstanarm default warmup of (floor(iter/2) and default uninformative priors (see[62]). As there was good chain mixing final models were fit using only one chain to obtain posterior samples for each step in the mortality pathway to use in a Monte–Carlo simulation to explore the impacts of thermal and suspended sediment related stress individually and as cumulative impacts. We focused on light as the primary suspended sediment related stressor because light was overwhelmingly the most important variable for bleaching in both groups of corals, the best predictor of mortality incidence in branching corals, but was also an important predictor of the amount of lost tissue in massive corals (Supplementary Table 2, Fig. 1). For the massive corals, where the incidence of mortality was strongly influenced by sedimentation, we estimated sedimentation pressure for each light level based on a Bayesian model fit of the observed water quality data (Supplementary Figure 5). In most cases we used the model with the lowest AICc (Supplementary Table 2) as our selected 'best' model. However, for the probability of live tissue loss in massive corals, we use the simplest model within 2 AICc (light + bleaching status), largely because this model included light rather than sedimentation, which was both more closely linked to the dredging stressor used in simulations, but also had a higher summed ωAICc importance score (Fig. 2).

For each iteration of the Monte–Carlo we predicted the incidence of bleaching, the incidence of mortality (which is dependent on bleaching status for both branching and massive corals), and the overall coral loss given mortality (also dependent on bleaching status). By multiplying the predicted probabilities for both the bleached and unbleached mortality pathways and then summing both pathways, we could calculate the total combined probability of coral loss under thermal pressure alone (high temperature and high light conditions in the absence of dredging; 31 °C and 8.8 mol photons m$^{-2}$ d$^{-1}$); dredging stressors alone (25 °C, range of light levels); and under cumulative thermal and dredging stress (high temperature [31 °C] across a range of light levels). A theoretical 'additive' curve was estimated as the sum of coral loss associated with thermal stress alone and that predicted for different levels of dredging stress at low temperature.

**Distance of impacts**. To provide context for the spatial scale of realised impacts of suspended sediments associated with dredging in the present study, as well as the potential scale of the shading effects that actually occurred, we estimated proportional coral loss as a function of distance from the dredging activities using the actual fortnightly water quality conditions observed at Barrow Island. We used predicted rather than observed total coral loss in order to partition the effects of dredging from that of thermal stress, and examined decay relationships with distance under high (31 °C) and low (25 °C) thermal stress scenarios. The data used were the observed worst case fortnightly values for light and sedimentation (see Supplementary Table 1) at each site, reflecting the data used in the original analyses on mortality (see 'Predicting coral mortality' above). As thermal stress events primarily occur during summer months (when light levels as measured through DLI are naturally higher[46]), our thermal stress scenario was based only on the data from the months of December, January and February. The total predicted fortnightly coral loss was modelled as a binomial function of starting coral cover (100 trials) against log(distance), where distance was the minimum distance to the dredging footprints as per[46]. We modelled distance on a log scale as previous analyses had indicated that water quality conditions change as an exponential function with distance from dredging[46]. Models were fit using stan_gamm4 with default priors and warmup, with 3 chains and 10,000 iterations. Site identifier and fortnight number were included as random effects to accommodate non-independence in time and space. An estimated 10% 'Effect Distance' (ED$_{50}$) was calculated as the distance at which the predicted curve drops below the 50% change from the maximum predicted value (at the closest observed distance of 0.19 km) to the minimum predicted value (at the farthest observed distance of 32.8 km, except for branching corals where the minimum occurred at intermediate distances from dredging, see Fig. 4).

**Reporting summary**. Further information on experiment design is available in the Nature Research Reporting Summary linked to this article.

## Data availability
The data that support the findings of this study are available through the Western Australian Marine Science Institution but restrictions apply to the availability of these data, which were used under license for the current study, and so are not publicly available. Data are however available from the corresponding author (R.F.) upon reasonable request and with permission of the Western Australian Marine Science Institution and Chevron Australia.

## Code availability
All R code used in the analyses presented are available at: https://github.com/AIMS/WAMSI-DSN-cumulative-impacts-bleaching-and-dredging

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

## Acknowledgements

This research was funded by the Western Australian Marine Science Institution (WAMSI) as part of the WAMSI Dredging Science Node, and made possible through investment from Chevron Australia, Woodside Energy Limited, BHP Billiton as environmental offsets and by co-investment from the WAMSI Joint Venture partners. Additional funding was supplied by a University Postgraduate Award to P.B.-B. This research was also enabled by data and information provided by Chevron Australia. The commercial entities had no role in data analysis, decision to publish, or preparation of the manuscript. The views expressed herein are those of the authors and not necessarily those of WAMSI.

## Author contributions

P.B.-B., R.J., and R.F. conceived the study. P.B.-B. analysed the images. P.B.-B. and R.F. conducted the statistical analysis. R.F. led the writing of the manuscript with P.B-B and R.J.

## Additional information

**Competing interests:** The authors declare no competing interests.

