## [Peer review file · Nature Communications]

Reviewer #1 (Remarks to the Author):

Review of “Synergistic and antagonistic impacts of suspended sediments and thermal stress on corals”

This manuscript analyzes data from 25 sites along a gradient of distance and impact from a single dredging location off the coast of Australia, to investigate the combined effects of dredging and temperature on coral bleaching, mortality, and tissue loss. The authors argue that there are “antagonistic” effects of temperature and dredging, when dredging effects are low to moderate.

I think this is an important topic, and the data, robustly analyzed and correctly interpreted, will likely be of interest to coral reef ecologists working on or otherwise interested in effects of climate or water quality stressors on reefs.

I think that the “antagonism” story is a bit oversold, given the analyses. Looking at Extended data Table 2, the lowest-AIC model only includes an interaction between a water quality parameter and temperature stress in the “probability of bleaching” model. For mortality incidence, the best-fit model for branching corals includes a light x bleaching interaction in the best-fitting model, but the next-best model is almost as good a fit ($\Delta AIC \sim 1$) and has no interaction between thermal stress and water quality variables. This suggests that any estimated interactive effects are going to be weak. For mortality incidence of massive corals and for proportional live tissue loss for both growth forms, there’s no evidence of an interaction.

Figure 2 similarly suggests that the only evidence of a biologically meaningful interaction is in the light-temperature interaction for bleaching probability. For mortality incidence, the rectangular-shaped contours for branching corals suggest that either high temperature, or low light, or both, are bad – moreso if you are bleached than if you are not. For massive corals, probability of mortality is driven mainly by sedimentation, and if anything there is a very slight suggestion of synergy (your mortality risk increases with sedimentation a bit earlier at high vs low temperatures). For colony loss, again, colony loss increases as light decreases for massive corals, both for bleached and unbleached corals.

Looking at the predictions in Figure 3, there doesn’t look to be strong evidence of departures from additivity in panel (a) (the confidence intervals on “Cumulative – Additive” look to encompass zero pretty much throughout the dredging stress range. In panel (b), the apparent increase in mortality as you go from about 3-4kms from dredging to 30kms (part of what is invoked as evidence in support of protective effects of low-to-medium dredging stress) is *very* small, relative to the breadth of the confidence region.

My own reading of these fits is that the data suggest that both dredging and bleaching are bad, in this system the effects of dredging are apparent only over the first 2 kms or so. The surprise is that the apparently protective effect of low light on bleaching incidence does not seem to translate into lower mortality incidence or tissue loss.

To sum up here, the management implication of someone just reading the conclusions might be that low to moderate dredging impacts might help mitigate the effects of bleaching stress. However, looking at the fitted models, and taking account of the very limited evidence of interactive effects on mortality incidence and tissue loss, I think a more straightforward reading of the results is that dredging is bad for branching corals above about 1.9 mean DLI, and massive coral mortality and tissue loss increases fairly monotonically with dredging stress. Temperature stress is also bad for corals, especially branching corals. High dredging stress plus temperature stress is especially bad.

With respect to methods, I am concerned about the use of repeated measures on the fitted models. I can appreciate that the authors have modelled these repeated measures as residual effects nested within colony-level effects, but there is a lot of temporal structure to those time series (and also the environmental time series at the site level) that is not accounted for in the model. Rather than run a really complicated model with autoregressive effects for time, the authors might consider extracting a single value for each response variable for each colony, and a single value for each explanatory variable for each site (or each logger), and then re-running the analysis. I'd feel better if I knew the results were the same from such an analysis. It would reassure me that the models weren't being over-fitted due to what would essentially be an inflated degrees of freedom problem.

Reviewer #2 (Remarks to the Author):

The manuscript 'Synergistic and antagonistic impacts of suspended sediments and thermal stress on corals', submitted to Nature Communications by Fisher et al, highlights the importance of considering multiple concurrent impacts on corals and coral reefs. In the case of their study, the authors have examined/wrangled a large dataset, gathered over ~18months, during a dredging project.

Overall, I think that the paper's claims are novel and will be of interest to a broad readership. It is certainly important to put a spotlight on the interactions of multiple coral and coral reef stressors, particularly ones that have the potential to be mitigated at local scales. I believe the methods to be sound, though I cannot comment specifically on the models used. I think that the paper should be published, but some further work is needed before hand.

The manuscript would benefit from careful proof reading (there are a number of typos e.g. check whole document for "morality" which should be mortality, and see below) and general tidying-up with regards to the language, to provide more clarity throughout. Expanding on the biological

meaning/discussion and interpretation of many of the results/figures would be helpful for the reader, particularly in how they relate to or are supported by cited work. While I appreciate that the manuscript is not a physiology paper, a little extra discussion on the costs of sedimentation / mucus production and how it affects overall coral health and tolerance may make the article more relevant and accessible to a wider readership.

Altering the coral categories to read simply Branching and Massive may help the flow of the paper. The first reference to them e.g. line 62, could read Branching (including reef building corals from the families Acroporiidae and Pocilloporiidae) and Massive (Poritidae), as you have more detail in the methods. See my notes below regarding the arguments detailed in the methods for your coral species selection.

Generally, I would suggest avoiding using the phrase 'tolerance to dredging', and similar, as the corals themselves are not dredged. It is the environmental shift (sedimentation & turbidity & smothering) that the coral is tolerant/susceptible to.

There are several sections, such as in the methods, where phrasing like "these corals" (line 340) is used. It would be helpful to the reader if the language was consistent and specific throughout the manuscript, and referred to the category (branching or massive).

The results and discussion section may benefit from a brief summary of the key findings at the beginning, to provide the reader with a framework to more easily follow the subsequent results and interpretation.

Further clarification and interpretation of results would be helpful, particularly for Fig 1. Line 62 – 64, for example, seems to suggest that everything drove mortality. It would be easier for the reader to follow if Fig 1 (a) and (b) were referred to in the text. I am slightly uncomfortable with the use of "bleached" as a thermal pressure metric for predicting mortality (Fig 1b) as it is a symptom of thermal pressure, rather than a type of thermal pressure (unlike temperature or light). However, I understand how including "bleached" is potentially helpful in the analysis and interpretation.

When referring to tissue loss, should partial mortality be referred to, because mortality would imply total tissue loss?

Citations 2 and 3 appear to be used to support the same argument, so I suggest choosing the most appropriate paper and removing the other, and similarly for citations 14 – 17. There is a great diversity of authors who have published excellent work on bleaching, the variability in coral tolerance (particularly among branching and massive growth forms) and physiological responses to sedimentation.

Methods

I believe that the case for selecting branching and massive coral species could be made much stronger and less complicated (more pithy). There are good life history strategy and winner/loser arguments, for example, that would support these groupings as well as their importance as abundant framework builders.

From line 247: Can you provide more information on the image analysis? Were the corals photographed with a scale or white balance? How was bleaching accurately categorised through time and among different coral species?

“These categories were converted to their equivalent proportional cover midpoints” – how was this done? Was each coral measured?

Minor comments and typos/suggestions

Line 38: hyphenate “climate change-induced”

Line 42: Remove “on” to read “sediments can impact corals” OR change to “sediments can have an impact on corals”

Line 49-51: I am unclear at this point whether the dredging project is the same as the project that included extensive monitoring, especially as line 62 starts with “Our results show...”

Line 52: hyphenate dredging-related exposure

Line 53: Can you add some information to qualify this comment: “This monitoring resulted in the largest dataset ever collected”, such as ‘...on coral reefs exposed to dredging

Line 54: hyphenate dredging-related

Line 49: Could a time frame be added here for the dredging project and bleaching event?

Line 83: correct “probably” to “probability”

Line 108: Can this phrasing be changed? I am not sure what live tissue loss means when talking about a dead coral.

Line 242: Correct to: Water temperature (C) was recorded using in situ loggers

Line 248: I am unclear what the 27-40 surveys are refereeing to – does this mean the photographs or were transects run? Please clarify

Line 270: add coral before families

Line 273: I would call massive and branching groups morphological, rather than taxonomic, especially as you have included two different coral families in the branching category.

Line 250: correct from to form.

Line 279 and in reference to branching species: I am not sure that “additional complexities in the interpretation of results” is a good reason for excluding data. Perhaps different phrasing would help here. At present it sounds like there might be more to the story, or a different story, had these corals been included. I think that there is a more solid case in considering morphological groups – see data from Lizard Island by Madin, Dornelas, Baird and Connolly about growth, fecundity and mortality for different growth forms, or simple Tolerant vs Susceptible reef building corals as they display strong physiological differences and responses to perturbations.

Line 282: delete relatively

.

Line 330: change morality to mortality

Line 639: Typo on x-axis of extended data Fig.5 “Light stresss”

Remove “Shown are” from figure legends.

I hope the suggested amendments do not prove too arduous!

Caroline Palmer

Reviewer #1 (Remarks to the Author):

Review of “Synergistic and antagonistic impacts of suspended sediments and thermal stress on corals”

This manuscript analyzes data from 25 sites along a gradient of distance and impact from a single dredging location off the coast of Australia, to investigate the combined effects of dredging and temperature on coral bleaching, mortality, and tissue loss. The authors argue that there are “antagonistic” effects of temperature and dredging, when dredging effects are low to moderate.

I think this is an important topic, and the data, robustly analyzed and correctly interpreted, will likely be of interest to coral reef ecologists working on or otherwise interested in effects of climate or water quality stressors on reefs.

I think that the “antagonism” story is a bit oversold, given the analyses. Looking at Extended data Table 2, the lowest-AIC model only includes an interaction between a water quality parameter and temperature stress in the “probability of bleaching” model. For mortality incidence, the best-fit model for branching corals includes a light x bleaching interaction in the best-fitting model, but the next-best model is almost as good a fit ($\Delta AIC \sim 1$) and has no interaction between thermal stress and water quality variables. This suggests that any estimated interactive effects are going to be weak. For mortality incidence of massive corals and for proportional live tissue loss for both growth forms, there’s no evidence of an interaction.

The “antagonism” story arises from the outcome of combining predictions from both the bleaching and mortality statistical models. The key interaction driving “antagonism” in branching corals is in fact the light and temperature stress in the “probability of bleaching” model, for which there is extremely high model support (AICc weight =1). That interaction, combined with the fact that “Bleached” is a key predictor appearing in the top model set of all the subsequent mortality models, suggests that the conclusion that shading alleviated some mortality in the corals at Barrow Island is quite robust. Interactions between thermal stress and sediment predictors may further increase the level of observed “antagonistic” effect, but their absence would not alter this main conclusion. That being said, we have in fact soften the “antagonism story” somewhat in the context of the discussion, to provide more balance between the “antagonism” and “synergistic” parts of the story (see more detailed comments below).

Figure 2 similarly suggests that the only evidence of a biologically meaningful interaction is in the light-temperature interaction for bleaching probability. For mortality incidence, the rectangular-shaped contours for branching corals suggest that either high temperature, or low light, or both, are bad – moreso if you are bleached than if you are not. For massive corals, probability of mortality is driven mainly by sedimentation, and if anything there is a very slight suggestion of synergy (your mortality risk increases with sedimentation a bit earlier at high vs low temperatures). For colony loss, again, colony loss increases as light decreases for massive corals, both for bleached and unbleached corals.

As above, the strong light-temperature interaction for bleaching probability is the key component supporting the “antagonistic” result. The other relationships the reviewer points out here are the key components leading to the “synergistic” impacts – i.e. that mortality is worse when severe dredging conditions are combined with bleaching. The main message of the analyses presented here is that both in fact occur, depending on the level of stress.

For mortality incidence in branching corals, we completely agree with the reviewer's interpretation, that: "... either high temperature, or low light, or both, are bad – more so if you are bleached than if you are not."

We have removed the text "only weakly related to light and temperature" to strengthen this point here, and added an additional sentence: "Overall, the greatest incidence of mortality of branching corals (both bleached and unbleached) occurred under conditions of both high temperature stress and extremely low light (Fig. 2b)". We also agree with the reviewer's interpretation regarding mortality in the massive corals, and have now captured this in the text: "... and was slightly higher with increased temperature". It is these patterns that lead to the strong "synergistic" impacts observed on our modelled mortality. There is no doubt that, for high dredging related pressure, the combined impacts of dredging and thermal stress are worse than would be expected in isolation. This point has been further strengthened in the discussion, with an additional sentence: "At high dredging related pressure our models suggest strong evidence for synergistic impacts between sediment related stress from dredging and thermal stress on corals, with overall mortality highest when these impacts occur together."

Looking at the predictions in Figure 3, there doesn't look to be strong evidence of departures from additivity in panel (a) (the confidence intervals on "Cumulative – Additive" look to encompass zero pretty much throughout the dredging stress range. In panel (b), the apparent increase in mortality as you go from about 3-4kms from dredging to 30kms (part of what is invoked as evidence in support of protective effects of low-to-medium dredging stress) is *very* small, relative to the breadth of the confidence region.

We respect the reviewer's point that our confidence intervals do overlap on the original Fig. 3, and that the actual evidence for "antagonistic" effects is weaker than the evidence for "synergistic" effects. However, the probability of antagonistic effects does reach ~80% for branching corals and is therefore substantial. We have made minor edited the results to emphasise that these patterns are not strong (e.g. using terms such as "appear slightly antagonistic" and "weak antagonistic effect"). We have also removed some of the text in the discussion around the relative distance of the "antagonistic" versus "synergistic" effects (see in more detail below).

My own reading of these fits is that the data suggest that both dredging and bleaching are bad, in this system the effects of dredging are apparent only over the first 2 kms or so. The surprise is that the apparently protective effect of low light on bleaching incidence does not seem to translate into lower mortality incidence or tissue loss.

We agree with the reviewer's statement "both dredging and bleaching are bad, in this system the effects of dredging are apparent only over the first 2 kms or so." We have added an additional sentence in the discussion to more clearly emphasis the "Synergistic" impacts of both stressors.

As discussed above, we feel that there is sufficient evidence that there are (albeit weak), antagonistic impacts related to shading. That light interacts with temperature stress to result in lower bleaching is well known, and we retain our discussion around this point (along with adding an additional relevant citation). However, we have removed discussion around the relative distance of the "antagonistic" versus "synergistic" effects, as we agree could potentially be interpreted as "a bit oversold".

To sum up here, the management implication of someone just reading the conclusions might be that low to moderate dredging impacts might help mitigate the effects of bleaching stress. However, looking at the fitted models, and taking account of the very limited evidence of interactive effects on

mortality incidence and tissue loss, I think a more straightforward reading of the results is that dredging is bad for branching corals above about 1.9 mean DLI, and massive coral mortality and tissue loss increases fairly monotonically with dredging stress. Temperature stress is also bad for corals, especially branching corals. High dredging stress plus temperature stress is especially bad.

While we are confident in our conclusions from the results presented in our analyses here, we agree with the reviewer that a certain level of moderation in our discussion is warranted to ensure that the management implications are not overstated. We have added a sentence in the latter part of the discussion more clearly articulating that the combined impacts of dredging and thermal stress are bad for coral: "At high dredging related pressure our models suggest strong evidence for synergistic impacts between sediment related stress from dredging and thermal stress on corals, with overall mortality highest when these impacts occur together". We have also deleted the sentence: "Indeed, for branching corals, the positive effects of shading during thermal stress can be observed up to distances of 15 km from the dredging activity (Fig. 3), which is in line with estimated upper limits of observed water quality changes associated with dredging at this location(Fisher et al. 2015)", as well as the sentence: "Thus at Barrow Island the positive outcome of shading associated with the dredging activities (the antagonistic cumulative effects) covered a much broader spatial area than any observed synergistic cumulative effects.". While these are interesting points they are not essential to the main take home message intended here, which is that impacts are not simply additive, and removal of both sentences results in a more moderate discussion that is less specific to Barrow Island. We have also changed reference to "suspended sediments" to "reduced light" in the final sentence: "Evidence for a shading effect associated with reduced light lends support to the idea that it may be possible to alleviate the coral loss associated with thermal stress through active mitigation efforts(Rau et al. 2012), at least at local scales."

With respect to methods, I am concerned about the use of repeated measures on the fitted models. I can appreciate that the authors have modelled these repeated measures as residual effects nested within colony-level effects, but there is a lot of temporal structure to those time series (and also the environmental time series at the site level) that is not accounted for in the model. Rather than run a really complicated model with autoregressive effects for time, the authors might consider extracting a single value for each response variable for each colony, and a single value for each explanatory variable for each site (or each logger), and then re-running the analysis. I'd feel better if I knew the results were the same from such an analysis. It would reassure me that the models weren't being over-fitted due to what would essentially be an inflated degrees of freedom problem.

We understand the reviewers concerns around the potential temporal dependence, given the data analysed are a time series. We have spent considerable effort on these analyses to ensure they are as robust as possible. We have explored temporal variograms for these data, which did not show strong consistent trends in the temporal autocorrelation that could be modelled effectively using the correlation structures available in mgcv. We are unable to use classic time series methods here (e.g. ACF) because these assume the time series are regular, which is not the case in this dataset because the exact time between surveys varies.

Our exploration of the data indicates that the temporal dependence is not strong. This is because, while the water quality data are originally daily, only a single data point is used from the time step between observations of the individual coral heads (1-3 weeks). Inclusion of a colony ID random effect should effectively deal with non-independence associated with the fact that some colonies are simply more robust or more prone to mortality and show low or high values throughout the time series.

Analysis at the scale of a single value for each site (such as mean or maximum) is not possible given the purpose of this paper, because it is the 'real' time analysis that allows examination of the actual relationship between the key predictors related to thermal stress and sediment stress. That is, we need to have a measure of both temperature and sediment pressure at the same time to explore these interactions. The fact that some sites may have had low light/high sediment at some time during the time series may be irrelevant if this did not occur during the period of thermal stress.

We have gone to considerable effort to avoid over fitting in our models including: restricting the "k" argument (the dimension of the basis used to represent the smooth term) in our gamm4 fits to ensure that the fitted smoothers are relatively monotonic – that is they broadly go "up" or "down"; restricting the number of predictors allowed to be included in any single model formulation to three or less; and excluding any models from the candidate set that contain highly correlated (Pearson correlation >0.4) predictors. We feel that these precautions have resulted in a sensible set of candidate models and resulting fits that are defensible given the inherent variability and characteristics of this dataset.

Reviewer #2 (Remarks to the Author):

The manuscript 'Synergistic and antagonistic impacts of suspended sediments and thermal stress on corals', submitted to Nature Communications by Fisher et al, highlights the importance of considering multiple concurrent impacts on corals and coral reefs. In the case of their study, the authors have examined/wrangled a large dataset, gathered over ~18months, during a dredging project.

Overall, I think that the paper's claims are novel and will be of interest to a broad readership. It is certainly important to put a spotlight on the interactions of multiple coral and coral reef stressors, particularly ones that have the potential to be mitigated at local scales. I believe the methods to be sound, though I cannot comment specifically on the models used. I think that the paper should be published, but some further work is needed before hand.

The manuscript would benefit from careful proof reading (there are a number of typos e.g. check whole document for "morality" which should be mortality, and see below) and general tidying-up with regards to the language, to provide more clarity throughout.

The manuscript has now been carefully proof read, and a range of typographical errors have been corrected, including correcting the miss-spelled 'morality'. The manuscript has been edited to improve clarity, in response to this comment, as well as many of the other reviewer comments (see more detailed comments below).

Expanding on the biological meaning/discussion and interpretation of many of the results/figures would be helpful for the reader, particularly in how they relate to or are supported by cited work. While I appreciate that the manuscript is not a physiology paper, a little extra discussion on the costs of sedimentation / mucus production and how it affects overall coral health and tolerance may make the article more relevant and accessible to a wider readership.

We have added some interpretation of the biological meaning of the results, as well as reference to the relevant literature. In particular, we have added an explanation and citations for why light loss is a strong predictor of coral mortality: "Periods of low light and darkness limit photosynthesis, which is considered detrimental as symbiont photosynthesis can provide corals with up to 90% of their daily

energy requirements (Muscatine 1990, Riegl and Branch 1995). Reduction in phototrophic energy generation potentially increases reliance on energy stores, which would gradually decline over time, resulting in mortality once sufficiently depleted (Anthony and Larcombe 2000)”; as well as sediment deposition: “While coral colonies employ a range of active and passive removal mechanisms to clear sediment (Stafford-Smith and Ormond 1992, Junjie et al. 2014, Duckworth et al. 2017), when deposition rates exceed these removal mechanisms, more rapid necrosis of tissue results (Philipp and Fabricius 2003, Weber et al. 2012).”

Altering the coral categories to read simply Branching and Massive may help the flow of the paper. The first reference to them e.g. line 62, could read Branching (including reef building corals from the families Acroporiidae and Pocilloporiidae) and Massive (Poritidae), as you have more detail in the methods. See my notes below regarding the arguments detailed in the methods for your coral species selection. *This has been done as suggested.*

Generally, I would suggest avoiding using the phrase ‘tolerance to dredging’, and similar, as the corals themselves are not dredged. It is the environmental shift (sedimentation & turbidity & smothering) that the coral is tolerant/susceptible to.

I was unable to find this phrase, or even the word ‘tolerance’ or ‘tolerant’ in a search of the manuscript.

There are several sections, such as in the methods, where phrasing like “these corals” (line 340) is used. It would be helpful to the reader if the language was consistent and specific throughout the manuscript, and referred to the category (branching or massive).

This has been checked throughout the manuscript and branching or massive corals now referred to explicitly where appropriate.

The results and discussion section may benefit from a brief summary of the key findings at the beginning, to provide the reader with a framework to more easily follow the subsequent results and interpretation.

We have now added both an additional summary figure and two brief summary paragraphs of text at the beginning of the results. This text more broadly describes the mortality and bleaching events observed during the dredging and co-occurring bleaching event at Barrow Island, which we think will help the reader follow the subsequent statistical results and interpretation.

Further clarification and interpretation of results would be helpful, particularly for Fig 1. Line 62 – 64, for example, seems to suggest that everything drove mortality. It would be easier for the reader to follow if Fig 1 (a) and (b) were referred to in the text.

These lines refer specifically to the results of the statistical analysis, particularly in terms of the key drivers. We now explicitly refer to parts a & b of the original Fig. 1, and also clarify that this figure relates to the statistical analysis. Addition of a new Fig. 1 describing more generally the mortality patterns should also aid in clarification and interpretation of the results.

I am slightly uncomfortable with the use of “bleached” as a thermal pressure metric for predicting mortality (Fig 1b) as it is a symptom of thermal pressure, rather than a type of thermal pressure (unlike temperature or light). However, I understand how including “bleached” is potentially helpful in the analysis and interpretation.

We agree with the reviewer that “bleached” is not necessarily a “thermal pressure metric”, but rather an indicator of “thermal stress”. The same logic also applies to “sediment cover”. These variables are a key part of the analysis, as also pointed out by the reviewer. To alleviate this criticism, we now explicitly refer to “suspended sediment and thermal pressure metrics and stress indicators” in the relevant table and figure captions, and clearly identify “bleached” and “sediment” cover as stress indicators in Extended data Table 1.

When referring to tissue loss, should partial mortality be referred to, because mortality would imply total tissue loss?

We now refer to mortality explicitly as “partial mortality” throughout. The addition of the new Fig. 1 and related text also deals more thoroughly with the concept of whole versus “partial” mortality, and provides some basic statistics for the rates of both types of mortality across the combined dredging and bleaching events that may be of interest to some readers.

Citations 2 and 3 appear to be used to support the same argument, so I suggest choosing the most appropriate paper and removing the other, and similarly for citations 14 – 17. There is a great diversity of authors who have published excellent work on bleaching, the variability in coral tolerance (particularly among branching and massive growth forms) and physiological responses to sedimentation.

Redundant references have been removed as suggested, with only the key ones retained.

Methods

I believe that the case for selecting branching and massive coral species could be made much stronger and less complicated (more pithy). There are good life history strategy and winner/loser arguments, for example, that would support these groupings as well as their importance as abundant framework builders.

We have strengthened the case for selecting the “branching” and “massive” growth forms, by including the text: “...representing two contrasting growth forms that likely differ in their susceptibility to both sedimentation and light stress.” We have also provided further clarification as well as justification regarding what, and why specific families/genera were included/excluded in these two groupings (see also more details below).

From line 247: Can you provide more information on the image analysis? Were the corals photographed with a scale or white balance? How was bleaching accurately categorised through time and among different coral species?

“These categories were converted to their equivalent proportional cover midpoints” – how was this done? Was each coral measured?

We have provided more information on the image analysis, as well as re-arranged the text to improve clarity here, as some of the requested information appeared later in the paragraph, and may have been missed. We have explicitly stated how the categories were converted to the equivalent proportional cover midpoints.

Minor comments and typos/suggestions

Line 38: hyphenate “climate change-induced” *Hyphen added as requested.*

Line 42: Remove “on” to read “sediments can impact corals” OR change to “sediments can have an impact on corals” “on” *has been removed as suggested.*

Line 49-51: I am unclear at this point whether the dredging project is the same as the project that included extensive monitoring, especially as line 62 starts with “Our results show...” *We agree with the reviewer that this may not have been clear, and have added text to explicitly state the dataset being analysed here are a direct result of the monitoring associated with dredging campaign. We have included the term “dredging” when referring to “The project” (originally lines 50 and 51 on the*

original manuscript), included additional information detailing that the monitoring data was collected as part of the dredging project, but before, during and after completion of the dredging activities.

Line 52: hyphenate dredging-related exposure *Hyphen added as requested.*

Line 53: Can you add some information to qualify this comment: “This monitoring resulted in the largest dataset ever collected”, such as ‘...on coral reefs exposed to dredging *Additional information has been added here to qualify this comment, as suggested.*

Line 54: hyphenate dredging-related *Hyphen added as requested.*

Line 49: Could a time frame be added here for the dredging project and bleaching event? *These time frame have been added.*

Line 83: correct “probably” to “probability” *This was corrected.*

Line 108: Can this phrasing be changed? I am not sure what live tissue loss means when talking about a dead coral.

This phrasing was changed to read: “When partial mortality events did occur, the proportion of live tissue lost was only weakly predicted by the pressure metrics examined”.

“Live tissue loss” is referring to the actual amount of partial mortality. We have now also clarified throughout that the “incidence of mortality” modelled in the first step of the analysis is partial mortality, rather than whole colony mortality.

In hindsight we realised that the difference between partial and full mortality of corals was not explicitly clear in the manuscript, and is very important from an ecological point of view. We have now included a new section at the start of the results more broadly summarising both the “whole” and “partial” colony mortality associated with the combined dredging and bleaching pressures, which may be of interest to some readers.

Line 242: Correct to: Water temperature (C) was recorded using in situ loggers *This was corrected.*

Line 248: I am unclear what the 27-40 surveys are refereeing to – does this mean the photographs or were transects run? Please clarify *These are separate field surveys undertaken to photograph the individual tagged colonies at each site. This has now been clarified in the text.*

Line 270: add coral before families *This has been added*

Line 273: I would call massive and branching groups morphological, rather than taxonomic, especially as you have included two different coral families in the branching category. *This has been changed as suggested where relevant*

Line 250: correct from to form. *This has been corrected*

Line 279 and in reference to branching species: I am not sure that “additional complexities in the interpretation of results” is a good reason for excluding data. Perhaps different phrasing would help here. At present it sounds like there might be more to the story, or a different story, had these corals been included. I think that there is a more solid case in considering morphological groups – see data from Lizard Island by Madin, Dornelas, Baird and Connolly about growth, fecundity and mortality for different growth forms, or simple Tolerant vs Susceptible reef building corals as they display strong physiological differences and responses to perturbations.

We agree with the reviewer that different phrasing was required to more carefully clarify why some coral families and/or genera were excluded from the analysis of both our “massive” and “branching” coral groups. This has now been rephrased in several places. We have included reference to Mirzerek, Baird and Madin (2018), as well as Darling et al. (2012). As suggested by the reviewer we have made a stronger case for the selection of contrasting “branching” and “massive” groups for analysis and provided further clarification regarding what, and why specific families/genera were included/excluded in our groupings.

We have taken considerable care with the available data to ensure that the two groups that we have analysed are consistently and well represented across the available study sites to ensure the results are robust.

Line 282: delete relatively. *We have deleted ‘relatively’, and also revised this sentence for clarity.*

Line 330: change morality to mortality *This has been corrected.*

Line 639: Typo on x-axis of extended data Fig.5 “Light stresss” *This has been corrected.*

Remove “Shown are” from figure legends. *This has been done as suggested.*

I hope the suggested amendments do not prove too arduous!

Caroline Palmer

Reviewer #1 (Remarks to the Author):

Overall, I think the authors have responded reasonably to the concerns I expressed in my original review. I have only a couple of remaining suggestions:

1. Given the shift in emphasis about the strength/magnitude of the antagonistic effects, it might be worth revisiting the wording of the relevant sentence in the summary paragraph/abstract, to ensure that it reflects the main text.
2. I think that a paragraph in the Methods summarizing the approach to checking model fit (including some of the checks the authors described in their response to reviewers document) is warranted and would be helpful to those readers like me who worry about whether such things have been done (and done well). I am not sure what length constraints the authors are under, but I hope there is sufficient flexibility to allow this.

Reviewer #2 (Remarks to the Author):

Having read the manuscript with the responses to reviewer comments, I think that the authors have improved the paper and addressed the concerns. As previously stated, I think that the results are interesting and informative for the study of coral reefs at a time when we urgently need to better understand the drivers and intricacies of mortality. I apologise for the delay in reviewing the revised manuscript.

I have some minor comments/changes that I noticed as reading through:

Line 67: Add "at" .. the end of the project

Line 69: delete "there were"

Line 79 – 82: is this a single sentence paragraph?

Line 91: change "in its own right" to independently?

Line 105: change to (mean daily temperature >29C)

109: consistent

218: unformatted citation in figure legend. There is a lot going on in this figure. Can "site" be added to the x axis labels, and it be made clearer the "branching" and "massive" are the names of the charts, not the axis labels? This perhaps could be clarified in the legend.

Caroline Palmer

Reviewers' Comments

Reviewer #1 (Remarks to the Author):

Overall, I think the authors have responded reasonably to the concerns I expressed in my original review. I have only a couple of remaining suggestions:

1. Given the shift in emphasis about the strength/magnitude of the antagonistic effects, it might be worth revisiting the wording of the relevant sentence in the summary paragraph/abstract, to ensure that it reflects the main text.

We have re-visited the abstract, and ensured that the relative weakness of antagonistic effects is clear, and reflects what is in the main text.

2. I think that a paragraph in the Methods summarizing the approach to checking model fit (including some of the checks the authors described in their response to reviewers document) is warranted and would be helpful to those readers like me who worry about whether such things have been done (and done well). I am not sure what length constraints the authors are under, but I hope there is sufficient flexibility to allow this.

We have added this paragraph as suggested.

Reviewer #2 (Remarks to the Author):

Having read the manuscript with the responses to reviewer comments, I think that the authors have improved the paper and addressed the concerns. As previously stated, I think that the results are interesting and informative for the study of coral reefs at a time when we urgently need to better understand the drivers and intricacies of mortality. I apologise for the delay in reviewing the revised manuscript.

I have some minor comments/changes that I noticed as reading through:

Line 67: Add "at" .. the end of the project

Added as suggested.

Line 69: delete "there were"

Deleted as suggested.

Line 79 – 82: is this a single sentence paragraph?

The sentence has been merged with the one above.

Line 91: change "in its own right" to independently?

This has been changed as suggested.

Line 105: change to (mean daily temperature >29C)

This has been changed as suggested.

109: consistent

This has been corrected.

218: unformatted citation in figure legend. There is a lot going on in this figure. Can "site" be added to the x axis labels, and it be made clearer the "branching" and "massive" are the names of the charts, not the axis labels? This perhaps could be clarified in the legend.

The unformatted citation has been corrected. Site has been added to the x-axis labels as suggested, with the headings for branching and massive moved up and to the side to try and help clarify they refer to the panel columns. That the left hand panels refer to "branching" and the right hand panels to "massive" is now also clearly stated in the figure caption. Note also that the colour scheme of this figures has been changed according to other editorial comments.